# Region Aware Transformer for Automatic Breast Ultrasound Tumor Segmentation

**Xiner Zhu**[1]                                          ZHUXINER@ZJU.EDU.CN
**Haoji Hu**[*1]                                          HAOJI_HU@ZJU.EDU.CN
**Hualiang Wang**[1]                                    HUALIANG_WANG@ZJU.EDU.CN
**Jincao Yao**[2,3]                                        YAOJC@ZJCC.ORG.CN
**Wei Li**[2,3]                                            LIWEI@ZJCC.ORG.CN
**Di Ou**[2,3]                                            OUDI@ZJCC.ORG.CN
**Dong Xu**[2,3]                                          XUDONG@ZJCC.ORG.CN

[1] *College of Information Science and Electronic Engineering, Zhejiang University, Hangzhou, China*
[2] *Cancer Hospital of the University of Chinese Academy of Sciences(Zhejiang Cancer Hospital), Hangzhou, China*
[3] *Institute of Basic Medicine and Cancer(IBMC), Chinese Academy of Sciences, Hangzhou, China*

## Abstract

Although Automatic Breast Ultrasound (ABUS) has become an important tool to detect breast cancer, computer-aided diagnosis requires accurate segmentation of tumors on ABUS. In this paper, we propose the Region Aware Transformer Network (RAT-Net) for tumor segmentation on ABUS images. RAT-Net incorporates region prior information of tumors into network design. The specially designed Region Aware Self-Attention Block (RASAB) and Region Aware Transformer Block (RATB) fuse the tumor region information into multi-scale features to obtain accurate segmentation. To the best of our knowledge, it is the first time that tumor region distributions are incorporated into network architectures for ABUS image segmentation. Experimental results on a dataset of 256 subjects (330 ABUS images each) show that RAT-Net outperforms other state-of-the-art methods.

**Keywords:** Automatic Breast Ultrasound, Medical Image Segmentation, Transformer, Self-Attention, Convolutional Neural Networks

## 1. Introduction

Breast cancer is one of the most common cancers in women (Bray et al., 2018). In Asia in 2018, it is estimated that there are almost $310,000$ breast cancer deaths and $911,000$ new cases (Zhang et al., 2021a). According to the clinic study, early identification of breast cancer can reduce the fatality rate by more than 40% (Huang et al., 2017; Horsch et al., 2002). The Automated Breast Ultrasound (ABUS) has been introduced for early detection of breast cancer to complement other medical imaging techniques such as magnetic resonance imaging (MRI), digital mammography and breast tomosynthesis (Lei et al., 2021). ABUS has been demonstrated to significantly improve detection accuracy of breast cancers. Studies have shown that ABUS is able to find up to 30% more cancers in women compared with mammogram (Drukker et al., 2002).

In addition to detection, breast tumor segmentation on ABUS images is of great interest both clinically and technically. From the clinical perspective, delineating tumor contours is

---

[*] Corresponding Author

vital for further assessing global and local tumor shapes, on which the most treatment choice criteria are based (Alemán-Flores et al., 2007). From the technical perspective, breast tumor segmentation on ABUS images is challenging and encounters difficulties of limited training data, shape and tissue variations, and small sizes of segmentation targets (Xian et al., 2018). For the medical image labeling work required by machine learning, the breast tumor segmentation is currently performed by manually tracing the lesion contours in most medical centers, which consumes a big amount of working time of medical doctors because ABUS produces a large number of images ($> 300$) per patient. Hence, effective and reproducible automatic segmentation methods are critical and necessary (Yang et al., 2019).

In this paper, we propose the Region Aware Transformer Network (RAT-Net) for breast tumor segmentation on ABUS images. We find that the problem of breast tumor segmentation on ABUS images has its intrinsic properties. First, the tumor regions follow strong prior distributions. Existing methods tend to neglect the prior region distribution of tumors, thus resulting in inaccurate segmentation (Hu et al., 2019; Zhou et al., 2021; Lei et al., 2021). Second, the sizes of tumors are tiny compared with the sizes of the ABUS images. As shown in Figure 4, the size of tumor regions is $\frac{1}{30} \sim \frac{1}{300}$ of the whole image. Thus, the attention mechanism needs to be specifically designed to incorporate much attention on tiny prior regions instead of equally attending the whole image. RAT-Net is specially designed based on the above two observations. The backbone of RAT-Net is UNet (Huang et al., 2020) with SegFormer as its encoder (Xie et al., 2021a). We incorporate region priors into the backbone to obtain the Region Aware Self-Attention Block (RASAB) and the Region Aware Transformer Block (RATB). Both blocks depend on calculating two region priors according to tumor distributions – (1) The suspicious region, in which at least one tumor has been occurred on all training images; (2) The high probability region, in which the frequency of tumors is above a threshold. We use the attention mechanism in Transformer (Vaswani et al., 2017) to calculate the query, key and value matrices to represent the two regions, which makes them interact with each other. The designed attention highlights the influence of regions with high tumor probability and decreases the influence of regions with low tumor probability, thus obtaining good segmentation performance and saving computation.

The contributions of our approach are as follows: (1) We propose RAT-Net, which is a region aware network architecture combining UNet with Transformer. To the best of our knowledge, this is the first time that tumor region distributions are incorporated into network architectures for ABUS tumor segmentation; (2) Extensive experiments and statistical significance tests demonstrate that our method achieves superior performance over most metrics compared with current state-of-the-arts in ABUS tumor segmentation.

## 2. Related Work

**Deep learning methods for semantic segmentation:** Semantic segmentation can be regarded as classifying each pixel in an image. Fully convolutional network (FCN) (Long et al., 2015) incorporates convolutional neural networks (CNN) for pixel-level classification. The UNet (Ronneberger et al., 2015) follows an encoder-decoder architecture in which long connections are devised to connect corresponding layers of the encoder and decoder. UNet structures are intensively implemented in medical image segmentation because of significant performance (Zhou et al., 2019; Diakogiannis et al., 2020; Li et al., 2018; Weng

et al., 2019; Huang et al., 2020). Another mainstream solution of semantic segmentation is the self-attention mechanism, which is firstly introduced by the non-local network (Wang et al., 2018). A lot of research is devoted to reducing the amount of computation for self-attention (Huang et al., 2019; Cao et al., 2019; Fu et al., 2019). The Transformer proposed by (Vaswani et al., 2017) can be regarded as an improvement of non-local network, which utilizes the query, key and value matrices to incorporate self-attention on cascaded stages.

**Transformer in medical image segmentation:** Recent research of medical image segmentation tends to apply Transformer to improve segmentation performance. (Zhang et al., 2021b) proposes MBTNet for corneal endothelial cell segmentation. TransUnet (Chen et al., 2021) is a combination of Transformer and UNet, which achieves convincing performance in abdominal CT image segmentation. (Valanarasu et al., 2021) proposes the medical Transformer which has two paths based on CNN and Transformer, respectively. (Hatamizadeh et al., 2021) introduces Transformers for 3D medical image segmentation, which utilizes a Transformer as the encoder to learn sequence representations of the input volume and effectively capture the global multi-scale information. CoTr (Xie et al., 2021b) efficiently joints CNN and Transformer for 3D medical image segmentation, in which CNN is constructed to extract feature representations and a deformable Transformer is built to model the long-range dependency on the extracted feature maps. Based on Swin Transformer (Liu et al., 2021), the Swin-Unet (Cao et al., 2021) is proposed for synapse multi-organ segmentation.

**Breast tumor segmentation for ABUS images:** There are several works discussing breast tumor segmentation for ABUS images in recent years. (Xian et al., 2015) combines space and frequency domains to automatically segment breast ultrasound images. (Hu et al., 2019) uses a dilated FCN combined with active contour to segment tumors in ABUS images. (Zhou et al., 2021) proposes a network modified by UNet to segment and classify tumors. (Lei et al., 2021) proposes the mask scoring R-CNN for breast tumor segmentation. It is rare in the current stage that region priors and Transformers are implemented into this area, which is the scope of this paper.

## 3. The Proposed Method

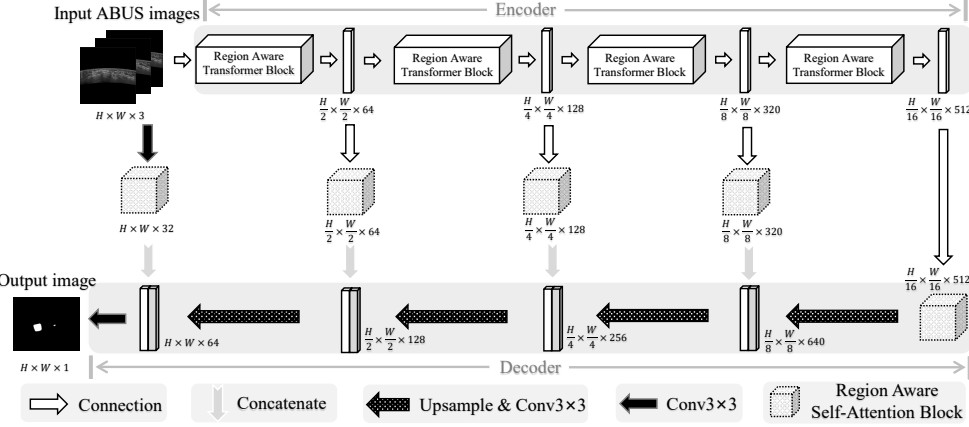

Figure 1: The overall architecture of the Region Aware Transformer Network (RAT-Net).

The overall architecture of the proposed Region Aware Transformer Network (RAT-Net) is shown in Figure 1. We input three consecutive images into RAT-Net. The output of RAT-Net is a predicted image consisting of tumor probability for each pixel. RAT-Net employs a typical encoder-decoder architecture, in which long connections are incorporated to connect corresponding layers within it. Both the encoder and the decoder have five stages. As for the encoder, we implement the SegFormer (Xie et al., 2021a), which is a specially designed Transformer block to improve performance. The Region Aware Self-Attention Blocks (RASAB) are added into the long connections to incorporate region prior information into the network. In addition, region prior information is also added into SegFormer through the Region Aware Transformer Block (RATB) to enhance performance. At the decoder side, the outputs of corresponding layers are concatenated and convolved at the stage level to generate segmentation results.

### 3.1. The Region Aware Component

The core element of RASAB and RATB is the Region Aware Component (RAC) as shown in Figure 2(a). Firstly, the input feature maps are segmented into two regions – (1) the suspicious region, in which at least one tumor has been occurred of all training images and (2) the high probability region, in which the frequency of tumors is above a threshold. Similar to the attention mechanism of Transformer, we use the query, key and value matrices to represent features of the two regions. By the Transformer-like attention, the RAC highlights the influence of regions with high tumor probability and reduces the influence of regions with low tumor probability.

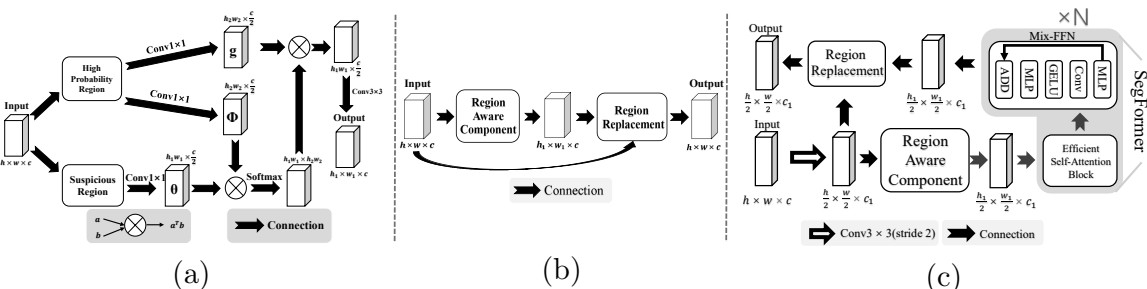

Figure 2: The architectures of the Region Aware Component (RAC).

**Segmentation of Suspicious and High Probability Regions:** The suspicious and high probability regions are calculated directly from the original ABUS images. We firstly count the frequency of tumors in all training samples, i.e., if tumor is occurred, the value corresponding to the pixel position of the tumor is increased by one. Then, we linearly normalize the heatmap to set the maximum frequency as 1 and no tumor as 0. Figure 3(a) shows the normalized heatmap where different colors from red to white represent different heatmap values from maximum to minimum, and the black color represents no tumor occurred at the corresponding position. It can be seen that tumors occur in a predefined small region of the whole image. Figure 3(b) depicts the region in which at least one tumor is occurred. The suspicious region is defined as the bounding rectangle of the white pixels in Figure 3(b), with $h_1$ and $w_1$ as its height and width respectively. Figure 3(c) is the heatmap

thresholded by a hyper-parameter $T \in [0,1]$ (here $T = 0.3$). The high probability region is defined as the bounding rectangle of the white pixels in Figure 3(c), with $h_2$ and $w_2$ as its height and width respectively. Supposing that the height and width of the original ABUS image is $H$ and $W$ respectively, we can obtain $h_1 = 0.24h$, $w_1 = 0.96w$, $h_2 = 0.13h$ and $w_2 = 0.53w$. After passing the image through convolutional layers of RAT-Net, the feature map size is shrinking gradually at each layer of the encoder and decoder. We keep the relative position of the suspicious and high probability regions on the feature map, and calculate $h_1$, $w_1$, $h_2$ and $w_2$ proportional to the original size of the ABUS image.

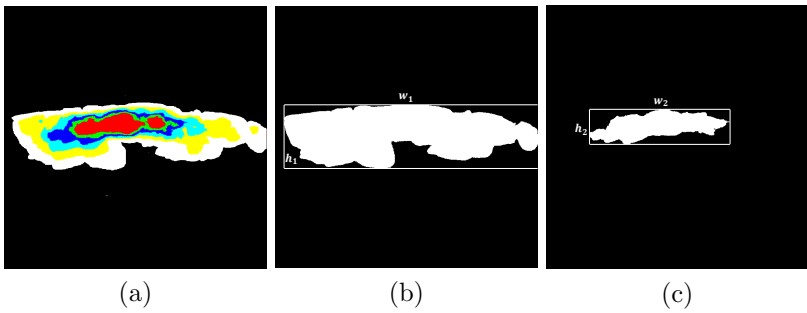

(a)                         (b)                         (c)

Figure 3: (a) The normalized heatmap of tumor frequency, in which we use red, green, blue, cyan, yellow and white colors to represent pixels with threshold $T$ greater than 0.5, 0.4, 0.3, 0.2, 0.1 and 0 respectively; (b) The suspicious region ($T > 0$); (c) The high probability region ($T > 0.3$).

**The Region Aware Component:** As shown in Figure 2(a), the output feature map of RAC keeps the same channel number as the input, but changes the height and width to $h_1$ and $w_1$, which are the height and width of the suspicious region. Three functions $g$, $\phi$ and $\theta$ are generated by $1 \times 1$ convolution. If we abbreviate the input of the suspicious region as $X_a$ and that of the high probability region as $X_b$, the output $Y$ is calculated as: $Y = Conv_{3 \times 3} \left[ Softmax \left[ \theta(X_a)\phi(X_b)^T \right]^T g(X_b) \right]$. This equation is similar to the self-attention of Transformer (Vaswani et al., 2017). Features of the two regions are interacted with each other by the same way as the query, key and value matrices in Transformer. By limiting the self-attention only in the suspicious and high probability regions, a big amount of computation is saved compared with attending the whole image.

### 3.2. The RASAB and RATB Blocks

Both the RASAB and RATB are based on RAC. As shown in Figure 2(b), RASAB obtains feature maps from the output of RAC, and substitutes part of the input by region replacement. Here region replacement means that we first replace the $h_1 \times w_1 \times c$ regions of the input with feature maps of RAC to obtain the new feature map and then concatenate the input feature map of RAC and the new feature map, and finally, we use a $3 \times 3$ convolution to adjust the number of output channels to be consistent with the number of input channels. The corresponding offset of the replacement is defined in Figure 3(b). Note that the

output of RASAB has the same size as the input, so RASAB can be regarded as a pluggable enhancement block to highlight the influence of regions with high tumor probabilities.

The architecture of RATB is shown in Figure 2(c). It can be seen from Figure 1 that RATB shrinks the input feature map by $\frac{1}{2}$, and also increases the channels by a predefined value. The input feature map is firstly convolved by $3 \times 3$ kernels with stride 2. After convolution, the feature map is fed into the RAC to enhance regions with high tumor probabilities. Then, the obtained feature map is fed into $N$ SegFormer basic blocks (Xie et al., 2021a) for further enhancement. Finally, region replacement takes place to obtain the output of RATB.

## 4. Experiments

We evaluate our method on the dataset of ABUS images taken from Cancer Hospital of the University of Chinese Academy of Sciences (Zhejiang Cancer Hospital), Hangzhou, China. A single ABUS scan of a breast generates 330 slices from top to bottom, each with a pixel size of $0.27 \times 0.27 \times 0.5$ mm. The range of tumor volume is $[0.024, 14.443]$ cc with an average and standard deviation of $3.011 \pm 4.433$ cc. Pathologically confirmed by needle biopsy or surgical resection. Institutional Review Board approval has been obtained. The dataset contains 256 subjects with 330 images each, and we randomly select 210 subjects for training and the remaining 46 subjects for testing. All images are labeled by three senior sonographers, who are the fifth, sixth and seventh authors of this paper, with 5-10 years of clinical work experience. The labeling work is double-blind, and the third doctor will judge when the labeling is controversial.

All our experiments are performed on four RTX3090. The optimizer is Adam (Kingma and Ba, 2014) and the learning rate is set to $10^{-4}$. The weight decay is $10^{-1}$ and the total number of training epochs is 100. For data pre-processing, we first resize the input images to $384 \times 384$, then augment them by randomly choosing several operations (Jung et al., 2020) from {GaussianBlur, Affine, Crop and Flip}. We use the accuracy (ACC), the 95th percentile of the asymmetric Hausdorff distance (HD95), Dice similarity coefficient (DSC) and mIoU as evaluation metrics. To compare the time-complexity of different methods, we utilize the number of parameters and FLOPs as indicators. Our code will be released in Github for a better understanding.

### 4.1. Hyperparameter Selection

For selecting the optimal threshold $T$, we randomly select 170 subjects from the training data to train the model, then we use the remaining 40 subjects as the validation set to decide the optimal $T$. As shown in Figure 3, a bigger value of $T$ produces a smaller high probability region, so the value of $T$ is important for selecting the high probability region. From Table 1, we can see that the performance of row 1-3 has its own merits, and this shows that our method is insensitive to the choice of T when $T$=0.1, 0.2 and 0.3. (Zhou et al., 2021; Lei et al., 2021) both use DSC and HD95 as metrics to evaluate ABUS segmentation, and when $T$=0.3, these two metrics reach the optimum, so we select $T$=0.3 as the hyperparameter for our next experiments. After the optimal $T$ is selected, we retrain the network with all the training data and test its performance on the test data.

Table 1: Hyperparameter tuning: different $T$ values.

|  | ACC (%) | HD95 (mm) | DSC (%) | mIoU (%) | Parameters | FLOPs |
|---|---|---|---|---|---|---|
| $T = 0.1$ | $84.43 \pm 13.57$ | $6.89 \pm 8.96$ | $68.23 \pm 31.45$ | $\mathbf{58.65 \pm 29.15}$ | 64.07M | 29.23G |
| $T = 0.2$ | $\mathbf{84.51 \pm 13.52}$ | $6.68 \pm 8.69$ | $68.33 \pm 30.37$ | $58.32 \pm 28.27$ | 64.07M | 29.21G |
| $T = 0.3$ | $84.32 \pm 13.51$ | $\mathbf{6.59 \pm 8.19}$ | $\mathbf{68.33 \pm 29.97}$ | $58.25 \pm 27.84$ | 64.07M | 29.21G |
| $T = 0.4$ | $84.26 \pm 13.74$ | $6.71 \pm 8.83$ | $67.49 \pm 32.31$ | $58.15 \pm 29.95$ | 64.07M | $\mathbf{29.20G}$ |
| $T = 0.5$ | $84.00 \pm 13.23$ | $6.76 \pm 8.54$ | $68.04 \pm 30.83$ | $58.20 \pm 28.79$ | 64.07M | $\mathbf{29.20G}$ |

## 4.2. Ablation Studies

Table 2: Three ablation studies: (1) Row 1-2: SegFormer with and without Region Aware Component (RAC); (2) Row 3-4: RAT-Net with and without the RASAB block.

|  | ACC (%) | HD95 (mm) | DSC (%) | mIoU (%) | Parameters | FLOPs |
|---|---|---|---|---|---|---|
| SegFormer | $76.43 \pm 13.17$ | $8.22 \pm 8.53$ | $53.76 \pm 37.62$ | $45.19 \pm 32.98$ | $\mathbf{63.99M}$ | 47.99G |
| SegFormer with RAC | $\mathbf{77.99 \pm 12.79}$ | $\mathbf{7.80 \pm 8.78}$ | $\mathbf{56.41 \pm 36.86}$ | $\mathbf{45.19 \pm 32.60}$ | $\mathbf{63.99M}$ | $\mathbf{47.52G}$ |
| RAT-Net_NORASAB | $79.98 \pm 11.41$ | $6.97 \pm 8.65$ | $62.09 \pm 34.61$ | $52.76 \pm 31.10$ | $\mathbf{62.51M}$ | $\mathbf{27.35G}$ |
| RAT-Net | $\mathbf{81.34 \pm 11.03}$ | $\mathbf{6.44 \pm 8.40}$ | $\mathbf{64.52 \pm 33.35}$ | $\mathbf{54.90 \pm 29.84}$ | 64.07M | 29.21G |

The first ablation study is to compare the SegFormer with and without the region aware component (RAC). For the experiment on SegFormer without RAC, only the grey block in Figure 2(c) is available. For another experiment, the whole block of Figure 2(c) is employed. Following settings of (Xie et al., 2021a), we set the block number $N = 3, 8, 27, 3$ for the four RATB blocks in Figure 1. Performance is exhibited in Row 5-6 of Table 2. SegFormer with RAC outperforms its counterpart over ACC, HD95, DSC and mIoU. It is counter-intuitive that the FLOPs is slightly decreased when RAC is added. The reason is that although RAC increases computation, it also produces feature maps with smaller sizes and thus decreases the computation of the SegFormer block.

The second ablation study is devised to evaluate the effectiveness of the RASAB block. RAT-Net_NORASAB represents that the RASAB is not added into the network, and RAT-Net means RASAB is added to all long connections. It can be concluded that RASAB significantly improves the segmentation performance while increasing computation overhead is almost negligible. Meanwhile, RASAB is a pluggable block and can be flexibly inserted into any place of the network.

## 4.3. Comparison with State-of-the-art Methods

We compare the proposed method with other state-of-the-arts including UNet, ResUNet, Swin-UNet, UTNet, TransUNet and SegFormer. Codes of these methods are publicly available and experimental settings are set the same. For fair comparisons, we set the number of training epochs to 100 for all methods, and the inputs of all contrast methods are three consecutive images. The results are shown in Table 3. A Student's t-test is performed between the results of the proposed method and each competing method with p-value listed

in Table 4. RAT-Net achieves the best performance over ACC, HD95, DSC, and mIoU in Table 3, and most of these comparisons showed statistical significance with p-values $< 0.05$. Table 5 indicates that our method is more sensitive to lesions than other methods. Please refer to Appendix A for more experiments and comments. Figure 4 qualitatively shows the results of three ABUS images. Methods for comparison are UNet, ResUNet, UTNet, TransUnet and RAT-Net. RAT-Net generates contours most consistent with the ground truth. Please refer to Appendix B for more qualitative segmentation results.

Table 3: Method comparison on the ABUS dataset.

| Method | ACC (%) | HD95 (mm) | DSC (%) | mIoU (%) | Parameters | FLOPs |
|---|---|---|---|---|---|---|
| UNet(Ronneberger et al., 2015) | $79.00 \pm 12.63$ | $7.96 \pm 9.09$ | $56.79 \pm 37.49$ | $48.37 \pm 33.54$ | 37.62M | 163.66G |
| ResUNet(Diakogiannis et al., 2020) | $79.95 \pm 12.05$ | $8.37 \pm 11.13$ | $57.94 \pm 36.55$ | $49.15 \pm 32.79$ | 70.79M | 163.66G |
| Swin-UNet(Cao et al., 2021) | $72.86 \pm 14.74$ | $11.37 \pm 12.81$ | $45.69 \pm 37.96$ | $37.60 \pm 32.71$ | 41.46M | 46.06G |
| UTNet(Gao et al., 2021) | $80.74 \pm 13.12$ | $7.60 \pm 10.39$ | $59.95 \pm 38.23$ | $52.16 \pm 34.48$ | **14.41M** | **25.37G** |
| TransUNet(Wang et al., 2021) | $80.50 \pm 11.54$ | $6.72 \pm 7.77$ | $62.41 \pm 35.23$ | $53.39 \pm 31.60$ | 105.57M | 72.30G |
| SegFormer(Xie et al., 2021a) | $76.43 \pm 13.17$ | $8.22 \pm 8.53$ | $53.76 \pm 37.62$ | $45.19 \pm 32.98$ | 63.99M | 47.99G |
| RAT-Net | **81.34±11.03** | **6.44±8.40** | **64.52±33.35** | **54.90±29.84** | 64.07M | 29.21G |

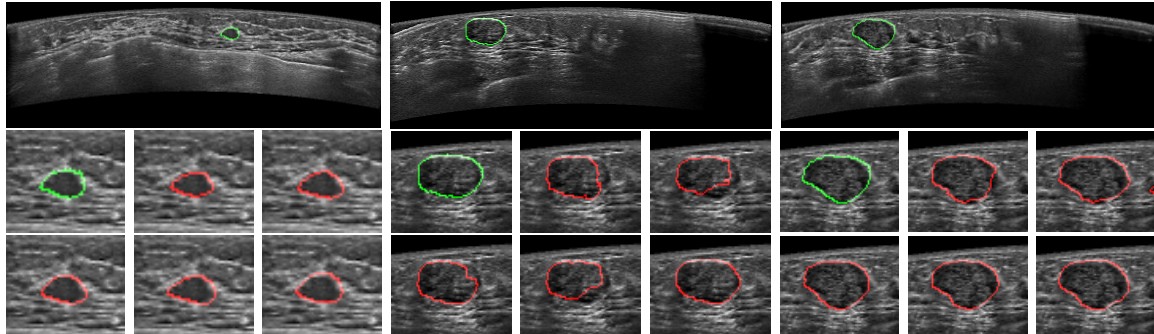

Figure 4: The qualitative segmentation results. The first row shows three ABUS images and their corresponding ground truths. Below each ABUS image, we show segmentation results of six methods, i.e., from top left to bottom right are the results of ground truth, UNet, ResUNet, UTNet, TransUnet and RAT-Net respectively.

## 5. Conclusions

We propose RAT-Net, a region aware Transformer-based method for Automatic Breast Ultrasound tumor segmentation. By incorporating region priors into the network architecture, we can effectively improve segmentation accuracy and reduce computation. We conduct experiments on a big dataset of ABUS images and find that RAT-Net outperforms many CNN- and Transformer-based methods. Although RAT-Net is designed for breast tumor segmentation on ABUS images, the region aware blocks can be embedded in different networks to improve performance and our method can be extended to other medical scenarios which have strong region prior distributions, such as brain and liver tumor segmentation.

## Acknowledgements

The study was supported in part by the National Natural Science Foundation of China (82071946 and U21B2004), the Natural Science Foundation of Zhejiang Province (LZY21F030001 and LSD19H180001) , the Medical and Health Research Project of Zhejiang Province (2021KY099 and 2022KY110), Zhejiang Provincial key RD Program of China (2021C01119) and the funds from the University Cancer Foundation via the Sister Institution Network Fund at the University of Texas MD Anderson Cancer Center.

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

## Appendix A. More experiments

Table 4: P-values of t-test between results of the proposed method and comparing method.

| Method | ACC | HD95(mm) | DSC | mIoU | Parameters | FLOPs |
|---|---|---|---|---|---|---|
| UNet(Ronneberger et al., 2015) | < 0.001 | < 0.001 | < 0.001 | < 0.001 | 37.62M | 163.66G |
| ResUNet(Diakogiannis et al., 2020) | 0.001 | < 0.001 | < 0.001 | < 0.001 | 70.79M | 163.66G |
| Swin-UNet(Cao et al., 2021) | < 0.001 | < 0.001 | < 0.001 | < 0.001 | 41.46M | 46.06G |
| UTNet(Gao et al., 2021) | 0.177 | 0.001 | < 0.001 | 0.011 | 14.41M | 25.37G |
| TransUNet(Wang et al., 2021) | 0.043 | 0.346 | 0.094 | 0.139 | 105.57M | 72.30G |
| SegFormer(Xie et al., 2021a) | < 0.001 | < 0.001 | < 0.001 | < 0.001 | 63.99M | 47.99G |

In Table 4, the p-values of most metrics are less than 0.05, which indicates that the proposed method significantly outperforms the corresponding competing method in individual evaluation metric. In the statistical significance test with TransUNet, three metrics are not lower than 0.05, but the advantage of our method lies in the smaller number of parameters and lower complexity.

Table 5: The sensitivity-specificity analysis.

| Metrics | UNet | ResUNet | Swin-UNet | UTNet | TransUNet | SegFormer | RAT-Net |
|---|---|---|---|---|---|---|---|
| **Sensitivity (%)** | $57.22 \pm 3.18$ | $55.67 \pm 2.99$ | $47.98 \pm 3.29$ | $60.62 \pm 3.16$ | $64.74 \pm 3.02$ | $54.15 \pm 3.19$ | $\mathbf{69.31 \pm 2.94}$ |
| **Specificity (%)** | $89.46 \pm 1.10$ | $\mathbf{92.25 \pm 0.82}$ | $87.34 \pm 1.49$ | $90.45 \pm 1.00$ | $87.24 \pm 1.20$ | $89.48 \pm 1.17$ | $84.92 \pm 1.25$ |

Table 5 compares the sensitivity and specificity results between the methods. As can be seen from Table 5, our method achieves the best performance in sensitivity, which indicates that our method is more sensitive to lesions and can better detect lesions. But our method does not perform well in specificity, which means that there are more areas that are not lesions that will be mistaken for lesions. However, for breast cancer diagnosis, it may be a good thing for the patient to be able to suggest more possible lesions during the computer-aided detection process, which means that the missed detection rate will be reduced. But the doctor will spend more time judging the marker area whether is a lesion.

Table 6: The experiment result of RAT-Net for detection (threshold=0.5).

| | Precision (%) | Recall (%) | F1 (%) | FPR (%) | TPR(Sensitivity) (%) | Specificity (%) |
|---|---|---|---|---|---|---|
| RAT-Net | $52.55 \pm 37.84$ | $76.05 \pm 35.18$ | $71.43 \pm 30.84$ | $0.64 \pm 1.37$ | $76.05 \pm 35.18$ | $99.19 \pm 4.27$ |

We partially modify RAT-Net to enable it to perform detection tasks and show its metrics on the ABUS dataset. Table 6 shows the result of RAT-Net for detection, and we set the threshold to 0.5. Compared to the sensitivity and specificity values shown in Table 5, both metrics are significantly improved when RAT-Net is used for detection tasks.

nnUNet(Isensee et al., 2021) is a deep learning-based segmentation method that automatically configures itself, including preprocessing, network architecture, training and post-processing for any new task. nnUNet will automatically perform fivefold cross-validation,

Table 7: The experiment result of nnUNet (3D).

|         | K1              | K2              | K3              | K4              | K5              | Mean            |
|---------|-----------------|-----------------|-----------------|-----------------|-----------------|-----------------|
| **DSC(%)** | $61.83 \pm 32.64$ | $59.16 \pm 31.86$ | $55.40 \pm 36.52$ | $54.67 \pm 31.07$ | $59.23 \pm 32.68$ | $58.06 \pm 32.95$ |

and the results of five experiments are shown in the Table 7. Compared with the standard UNet in Table 3, the performance of nnUNet in DSC is greatly improved. Since the ABUS dataset has 3D characteristics, turning the input of the dataset into 3D may better extract the characteristics of the dataset.

## Appendix B. Segmentation results of ABUS dataset

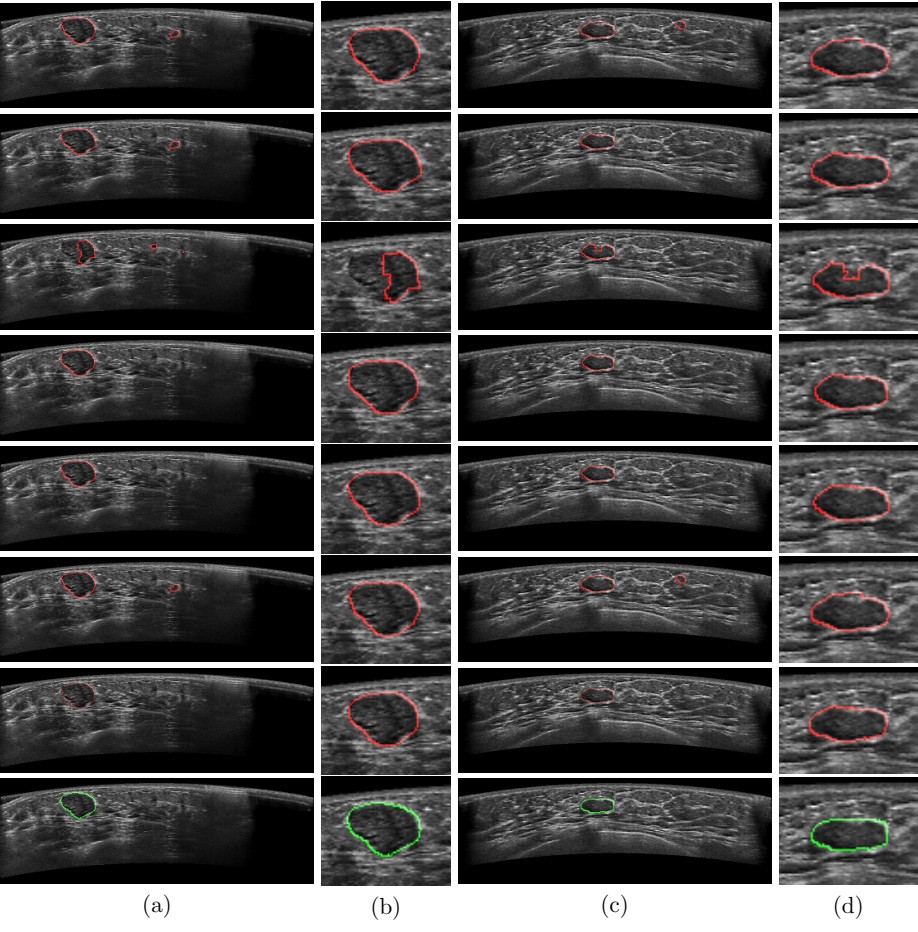

(a)          (b)          (c)          (d)

Figure 5: Segmentation results of two ABUS images. Column (a) and (c) are the whole images while Column (b) and (d) are the enlarged images around the tumor regions. Row 1-8 are segmentation results of Unet, ResUnet, Swin-Unet, UTNet, TransUnet, SegFormer, the proposed RAT-Net and ground truth respectively.

## Appendix C. Some image displays during training and testing.

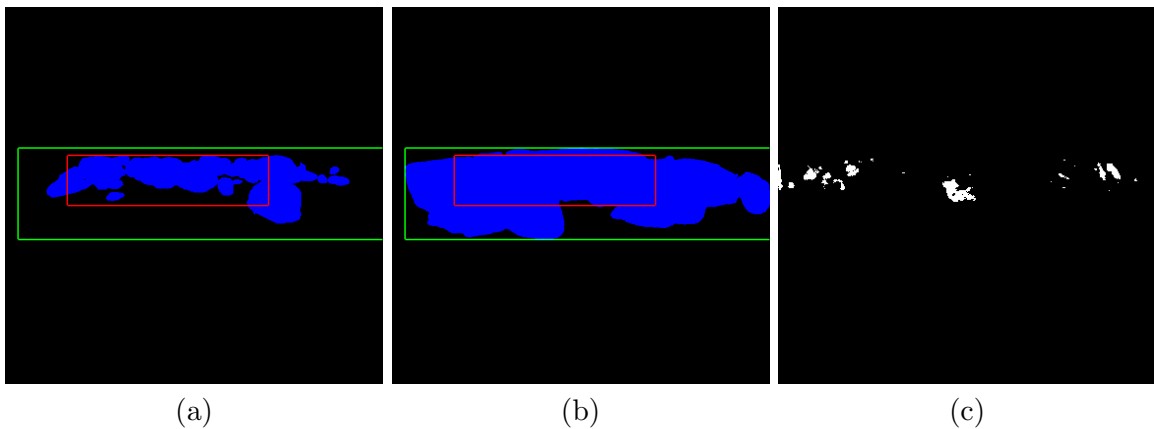

(a)           (b)           (c)

Figure 6: Green boxes represent Suspicious Region and red boxes represent High Probability Region. (a) a visualization of the tumor location in the test set; (b) a visualization of the tumor location in the training set; (c) Output image display when training is insufficient.

The Figure 6(a) shows a visualization of the tumor location in the test set. Note that, however, this is only for discussion of results after all experiments have been completed. All prior regions are obtained only from the information of the training set.

During the model learning, due to insufficient training at the beginning, different regions in the output image may have different accuracy. As shown in Figure 6(c), the segmentation accuracy of the prior region of the image will be higher than that of other regions. However, as the training continues, these seemingly obtrusive boundaries will gradually be eliminated, and finally become a smooth output. This is because not in the two prior regions does not mean that no features are extracted. Please see: 1) the long connection on the first left in Figure 1. Before the feature map enters RASAB, feature extraction has been performed through the $3 \times 3$ convolution. The number of channels has become 32, that is, feature extraction is still performed by convolution where not identified as a prior region, and then it enters RASAB to further calculate the area counted by the training set so that the target in the prior area can be better detected. Ablation experiment 2 in Table 2 also proves that the RASAB we added can detect the target better with a small increase in the amount of calculation; 2) In Figure 2(c), before entering the RAC block, the feature map will be extracted by $3 \times 3$ convolution with stride 2, and finally the region replacement will be performed; 3) The decoder part operates globally, and can also extract features regardless of whether it is a prior region.

