# OpenReview forum: "Region Aware Transformer for Automatic Breast Ultrasound Tumor Segmentation"
_MIDL.io/2022/Conference — MIDL 2022_

### Official Review · Reviewer_YcoZ · 2022-01-24

**Confidence:** 4
**Preliminary Rating:** 1

**Summary:**

The authors present a transformer-based approach for lesion segmentation in automated breast ultrasound (ABUS) images.  The approach is based on the SegFormer method, extended with "region aware components".  In essence, these region aware components are box ROIs in which an extra attention layer is applied.  The location and size of the ROIs is not learned, but derived from the data.

**Strengths:**

The method is based on SoA technologies and the dataset is large and interesting (albeit private).  The literature overview is good.  The authors have performed a few relevant ablation studies (although the evaluation has a few weaknesses).

**Weaknesses:**

A severe problem with the evaluation is that apparently, hyperparameter tuning was performed on the test set.  What the authors call "the first ablation study" is in fact not an ablation study, but hyperparameter tuning, and from Table 1, it appears that this was performed on the same test set that the final results are reported on.  The text also does not mention otherwise.  Unfortunately, this means that the observed "improvements" could be a matter of chance, and a new test set (that is not used for another round of tuning) would be required to prove otherwise.

Furthermore, the main contribution is of questionable quality: Region priors should better be learned, not hand-crafted. The proposed approach is just taking bounding boxes of regions from the training set (that is, one would hope that the analysis was performed on the training set only, because this is not described).

I am not too familiar with ABUS, so I will give the authors the benefit of the doubt concerning the question whether region priors can make sense, or whether positioning variability makes this dangerous anyhow.  Given my general US knowledge and the image appearance, though, I would expect a good deep learning setup, such as a transformer-based architecture, to be able to learn such priors from the images themselves, in which the imaging boundaries can be seen.  That approach would appear much more robust to me.

No statistical significance tests were mentioned.

**Deanonymize Review:**

yes

**Detailed Comments:**

You highlight that RASAB is a pluggable component, because it generates the same *size* as the input (which is actually achieved by placing the smaller ROI output within a copy of the input).  Is there any good argument for why the output for the ROI and the background should have "compatible" features?  If you rely on the model to just learn to produce these, have you tried to check that this works (by inspecting the RASAB output for ROI borders, either manually or automatically)?

Is it relevant to mention that the experiments were run on a server with four RTX3090?  Please specify whether the experiments were using multiple GPUs, otherwise just mention the GPU they were run on.

Semantic segmentation is indeed voxel classification, but why is that an "improvement" over image classification?  (IMO it's related, but just a different task. People interested in image classification would not necessarily find that segmentation models bring only improvements.)

I suggest to use small spaces (\,) before units (cc,mm), but no space after the thousands-comma.

**Final Rating After The Rebuttal:**

3: Borderline

**Justification Of The Final Rating:**

I have raised my rating to "weak reject", because I definitely want to acknowledge the work the authors put into the rebuttal phase. The authors addressed my raised issues and adapted the paper accordingly as good as possible under these circumstances. Still, the text itself could not convince me that the method development followed best scientific practices, and I believe the presented measures are overestimated due to overfitting in the whole experimental process. I realize that I am the most critical reviewer in this case, and I still wish the authors success both with their scientific work as well as with applying their method to other ABUS datasets.

**Paper Type:**

both

**Questions To Address In The Rebuttal:**

When you wrote that "RAT-Net achieves the best performance over ACC, HD95, DSC, and mIoU, surprisingly outperforming the second-best method by a large margin, while consuming the relatively small computation.", I wonder which "large margin" you mean. I would personally think that mIoU is the most relevant measure here, where RAT-Net scores 54.9+-30, while TransUNet scores 53.39+-32.

Please elaborate on any dataset splits you performed, and which steps of the analysis, training, hyperparameter tuning, and model selection was performed on which subset of the data.  Ideally, only the final(!) evaluation would have been performed on the test set.

**Special Issue:**

no

---

### Meta-Review · Area_Chair_obAi · 2022-02-20

**Recommendation:** Accept (Poster)
**Confidence:** 4

**Metareview:**

This paper describes an architecture to segment tumors in breast ultrasound images. The authors have responded well to the comments and concerns raised by the reviewers and made subtantial improvements to the paper. The evaluation problem on the test set has been partly addressed retroactively.

---

### Decision · Program_Chairs · 2022-02-28

Accept